# Environmental Enrichment Prevents Gut Dysbiosis Progression and Enhances Glucose Metabolism in High-Fat Diet-Induced Obese Mice

**DOI:** 10.3390/ijms25136904

**Published:** 2024-06-24

**Authors:** Rubiceli Manzo, Luigui Gallardo-Becerra, Sol Díaz de León-Guerrero, Tomas Villaseñor, Fernanda Cornejo-Granados, Jonathan Salazar-León, Adrian Ochoa-Leyva, Gustavo Pedraza-Alva, Leonor Pérez-Martínez

**Affiliations:** 1Laboratorio de Neuroinmunobiología, Departamento de Medicina Molecular y Bioprocesos, Instituto de Biotecnología, Universidad Nacional Autónoma de México (UNAM), Cuernavaca 62210, Morelos, Mexico; rubiceli.manzo@ibt.unam.mx (R.M.); maria.diaz@ibt.unam.mx (S.D.d.L.-G.); tomas.villasenor@ibt.unam.mx (T.V.); jsalazar@ottawaheart.ca (J.S.-L.); gustavo.pedraza@ibt.unam.mx (G.P.-A.); 2Departamento de Microbiología Molecular, Instituto de Biotecnología, Universidad Nacional Autónoma de México (UNAM), Cuernavaca 62210, Morelos, Mexico; luigui.gallardo@ibt.unam.mx (L.G.-B.); fernanda.cornejo@ibt.unam.mx (F.C.-G.); adrian.ochoa@ibt.unam.mx (A.O.-L.)

**Keywords:** enriched environment, inflammation, dysbiosis, colon, epithelial barrier, goblet cells, gut microbiota, hypothalamus, energy balance

## Abstract

Obesity is a global health concern implicated in numerous chronic degenerative diseases, including type 2 diabetes, dyslipidemia, and neurodegenerative disorders. It is characterized by chronic low-grade inflammation, gut microbiota dysbiosis, insulin resistance, glucose intolerance, and lipid metabolism disturbances. Here, we investigated the therapeutic potential of environmental enrichment (EE) to prevent the progression of gut dysbiosis in mice with high-fat diet (HFD)-induced metabolic syndrome. C57BL/6 male mice with obesity and metabolic syndrome, continuously fed with an HFD, were exposed to EE. We analyzed the gut microbiota of the mice by sequencing the 16s rRNA gene at different intervals, including on day 0 and 12 and 24 weeks after EE exposure. Fasting glucose levels, glucose tolerance, insulin resistance, food intake, weight gain, lipid profile, hepatic steatosis, and inflammatory mediators were evaluated in serum, adipose tissue, and the colon. We demonstrate that EE intervention prevents the progression of HFD-induced dysbiosis, reducing taxa associated with metabolic syndrome (*Tepidimicrobium*, *Acidaminobacteraceae*, and *Fusibacter*) while promoting those linked to healthy physiology (*Syntrophococcus sucrumutans*, *Dehalobacterium*, *Prevotella*, and *Butyricimonas*). Furthermore, EE enhances intestinal barrier integrity, increases mucin-producing goblet cell population, and upregulates Muc2 expression in the colon. These alterations correlate with reduced systemic lipopolysaccharide levels and attenuated colon inflammation, resulting in normalized glucose metabolism, diminished adipose tissue inflammation, reduced liver steatosis, improved lipid profiles, and a significant reduction in body weight gain despite mice’s continued HFD consumption. Our findings highlight EE as a promising anti-inflammatory strategy for managing obesity-related metabolic dysregulation and suggest its potential in developing probiotics targeting EE-modulated microbial taxa.

## 1. Introduction

Obesity is defined by the WHO as abnormal or excessive fat accumulation that presents a risk to health and is considered a global health epidemic. As of 2016, 39% of the world’s adult population was overweight, with 13% classified as obese. The swift rise in global obesity rates has prompted substantial research attention due to its profound implications. Obesity is a multifaceted condition that contributes to the development of chronic–degenerative diseases, including cancer, type 2 diabetes (T2D), metabolic syndrome, cardiovascular diseases, and neurodegenerative diseases [1,2]. One of its primary causes is consuming a Western diet (or high-fat diet (HFD)) [3]. Overnutrition and obesity induce chronic, low-grade, yet persistent inflammation (local and systemic) termed meta-inflammation that affects various organs such as the liver, pancreas, intestine, and brain [3,4]. Meta-inflammation significantly impairs glucose metabolism, disrupts the lipid profile characterized by elevated serum cholesterol and triglyceride levels, and leads to insulin and leptin resistance favoring the development of degenerative diseases [1,5,6]. In the liver, obesity increases lipid accumulation and activates various inflammatory pathways that contribute to insulin resistance and dysregulation of both glucose and lipid metabolism [7]. In adipose tissue, excess nutrient intake generates cellular stress, inducing the expression of cytokines and chemokines, which leads to the recruitment and activation of various immune cells, primarily macrophages [8]. This inflammatory process inhibits catecholamine and insulin signaling, thereby impairing normal adipose tissue function [9]. Numerous studies have demonstrated that inflammation is the key factor in impairing insulin signaling [10].

Notably, obesity is also associated with dysbiosis, a disturbance in the composition of the gut microbiota [11,12,13,14,15]. Proof of the causative role of gut microbes in influencing host metabolism was provided by the observation that germ-free mice do not develop obesity when fed a HFD. However, when feces from obese mice is transplanted into germ-free mice, the disease develops [16,17,18]. Changes in the microbiota composition of obese compared to lean individuals have been reported in both animal and human studies. Interestingly, these studies suggest an inverse correlation between the abundance of specific bacteria and adiposity, diabetes, and meta-inflammation. This indicates that changes in the microbiome contribute to obesity.

Murine models and cross-sectional human studies have shown that changes in the diversity and relative abundances of the microbiota and its metabolites are associated with the development of various pathologies, including Alzheimer’s disease, Parkinson’s disease, autism, depression, anxiety, gastrointestinal diseases, and obesity [19,20,21,22,23]. Currently, intense research is focused on identifying strategies to modulate gut microbiota composition for human health.

Environmental enrichment (EE) promotes social, visual, motor, cognitive, and somatosensory stimulation [24]. EE has shown promising effects in the context of obesity, both in preventing and reversing glucose metabolism disorders resulting from diet. In a mouse model of diet-induced obesity, EE effectively prevents weight gain and promotes adipose browning by elevating hypothalamic brain-derived neurotrophic factor (BDNF) levels [25]. Furthermore, EE exerts therapeutic effects on metabolically compromised mice by reducing inflammation in adipose tissue and the hypothalamus, thereby restoring insulin signaling [26]. Moreover, EE’s beneficial impact extends to murine models of colitis, where it attenuates experimental colitis by decreasing inflammatory markers in the colon. Accordingly, EE has the potential to modulate the intestinal microbiota and alleviate inflammation in a mouse model of Parkinson’s disease [27]. In this context, EE not only reduces disease markers but also increases bacterial taxa associated with a healthy state, such as the *Streptomyces* genus and *Macellibacteroides*, while decreasing those linked to disease, such as the *Desulfovibrio* and *Alistipes* genera [22,27,28,29,30].

Here, we investigated whether EE could prevent the progression of gut dysbiosis in obese mice that previously presented with metabolic alterations and that were continuously fed a HFD. We uncovered that EE demonstrates therapeutic efficacy in obese mice subjected to a HFD by decreasing taxa associated with metabolic syndrome and glucose metabolic disorders (*Fusibacter*, *Tepidimicrobium*, and *Acidaminobacteraceae*) while concurrently promoting the abundance of beneficial intestinal taxa (*Syntrophococcus sucrumutans*, *Dehalobacterium*, *Prevotella*, and *Butyricimonas*). This is concomitant with the restoration of glucose metabolism and a reduction in adipose tissue inflammation, liver steatosis, cholesterol, and HDL levels. Notably, obese mice exhibited lower circulating levels of lipopolysaccharides (LPS) despite being maintained on a HFD. Additionally, we observed an increased number of mucus-producing goblet cells in the colon and elevated levels of the mucin Muc2, indicating the positive impact of the EE on the intestinal barrier function in obese mice fed with a HFD.

In addition to its promising therapeutic applications in clinical settings for conditions such as neurodevelopmental disorders, age-related cognitive decline, neurodegenerative diseases, and chronic pain [31,32,33,34], our current findings suggest that exposure to an EE could serve as a lifestyle intervention to mitigate the adverse health effects of metabolic syndrome. Nonetheless, further research is necessary to confirm this hypothesis in humans.

## 2. Results

### 2.1. Environmental Enrichment Reverses the Metabolic Syndrome Induced by a High-Fat Diet Consumption

Initially, C57BL/6 mice (3–4 weeks old) were subjected to gut microbiota homogenization to eliminate compositional differences. During this time, the animals were fed a regular chow diet (ND) for four weeks (Appendix A). As expected, the ND-fed animals had normal glucose tolerance and insulin sensitivity (Appendix A). Likewise, 16S rRNA sequencing confirmed microbiota homogenization since no difference in the microbiota composition among individual mice was observed (Appendix A). Following microbiota homogenization, the mice were housed in standard conditions (NE) at week 0 and were divided into two groups: one group was fed a ND, while the other group was fed a HFD for 12 weeks (Figure 1A). As expected [26], compared to the ND-fed group, mice on the HFD exhibited increased weight gain (Figure 1B), higher weekly caloric intake (Figure 1C), elevated fasting blood glucose levels (Figure 1D), glucose intolerance (Figure 1E,F), insulin resistance (Figure 1G,H), increased total cholesterol (Appendix A), and low-density lipoprotein (LDL) cholesterol (Appendix A) levels in serum without changes in high-density lipoprotein (HDL) cholesterol (Appendix A) or triglyceride levels (Appendix A). We also observed increased lipid accumulation in the liver (Appendix A). In white adipose tissue (WAT), HFD-fed mice exhibited larger adipocytes (Appendix A), a higher number of infiltrating cells (Appendix A), and a greater area of infiltration (Appendix A) compared to ND-fed mice. Additionally, we observed increased damage to colonic crypts and epithelial cells (Figure 1I) and a reduction in goblet cell area (Figure 1J,K). Furthermore, the HFD led to gut dysbiosis, as evidenced by reduced bacterial diversity (Figure 1L,M) and significant alterations in the composition of the microbiota (Figure 1N,O), as previously reported [12,35,36,37,38]. Indeed, the ANOSIM confirmed a significant change in the microbiota caused by diet (R = 0.9630, *p* = 0.002) (Figure 1N).

Once our diet-induced obesity model was validated, the mice were divided again, with some remaining in standard housing conditions (ND NE and HFD NE), while others were housed in an environmental enrichment condition (ND EE and HFD EE), maintaining the same diet they had for the previous 12 weeks (Appendix A). After the switch in housing conditions, mice were further maintained for an additional 12 weeks (Appendix A). Notably, for the ND mice, we observed no significant differences in weight or other metabolic parameters between those housed in either a NE or an EE (Appendix A). Conversely, in the case of HFD-fed mice, we noticed that those in the HFD NE group continued gaining weight throughout the experiment (Appendix A), while HFD-fed mice housed in an EE lost weight, almost reaching the weight of ND-fed mice (Appendix A). This weight loss observed in the HFD EE mice was accompanied by a reduction in their weekly caloric intake (Appendix A). The EE also significantly improved glucose metabolism, reducing fasting glucose levels (Appendix A), enhancing glucose tolerance (Appendix A), and increasing insulin sensitivity (Appendix A) in the HFD-fed mice, as we previously reported [26]. Furthermore, the EE (HFD EE) effectively prevented the HFD-induced increase in total cholesterol (HFD NE; Figure 2A) and HDL (HFD NE; Figure 2B) serum levels, though it did not reduce LDL levels (Figure 2C). In our experimental conditions, no differences were found in triglyceride levels (Figure 2D). However, we did observe a notable increase in hepatic lipid accumulation in mice exposed to a HFD for 24 weeks (Figure 2E; HFD NE). Interestingly, the EE effectively prevented the development of hepatic steatosis in HFD-fed mice (Figure 2E; HFD EE). Moreover, the EE mitigated the inflammation caused by the HFD to the white adipose tissue (Figure 2F), resulting in fewer infiltrating cells (Figure 2G) and a reduced area of infiltration (Figure 2H) in the HFD EE mice compared to the HFD NE mice. Additionally, HFD-fed mice exhibited higher levels of IL-1β (Figure 2I) and IL-6 (Figure 2J) in WAT than ND-fed mice. However, the EE reduced IL-1β levels (Figure 2I) in WAT. A similar trend was observed for IL-6 and TNF levels in WAT (Figure 2J,K). These results confirm that EE significantly enhances glucose metabolism and alleviates inflammation in adipose tissue in HFD-fed mice, as previously reported [26].

### 2.2. Protective Effects of Environmental Enrichment on Metabolism Correlates with Mitigating High-Fat Diet-Induced Dysbiosis

As HFD-induced obesity is linked to gut microbiota alterations (dysbiosis) [12,37,38], and EE has the potential to modulate the intestinal microbiota and alleviate inflammation in a mouse model of Parkinson’s disease [27], we investigated whether EE provides metabolic protection for obese mice fed a HFD by modulating gut microbiota under our experimental conditions. We observed that the HFD led to a reduction in diversity (Shannon index) (Figure 3A), richness (observed operational taxonomic units or OTUs) (Figure 3B), and changes in bacterial composition (Figure 3C) compared to ND-fed mice (ND NE group). However, our housing conditions did not significantly affect bacterial diversity and richness among mice fed the same diet (Figure 3A,B). Interestingly, beta diversity analysis revealed a clear distinction between ND and HFD-fed mice (Figure 3D). While all ND-fed mice clustered together, different populations were observed in HFD-fed mice, with samples from the HFD NE and HFD EE groups showing a clear separation (Figure 3D).

To understand how fecal bacterial composition evolves during obesity progression, we compared bacterial taxa in mice fed a HFD and housed in standard conditions for either 12 (HFD) or 24 weeks (HFD NE) (Appendix A). This revealed that certain bacterial groups enriched in the HFD group housed in standard conditions for 12 weeks (HFD group) were diminished after 12 additional weeks on the HFD while others increased (HFD NE group) (Appendix A). This demonstrates that extended HFD exposure results in further changes in bacterial composition. As expected, beta diversity analysis revealed that the mice fed a HFD for 12 weeks (HFD) did not overlap with the HFD NE group (Figure 3D).

We then compared HFD-fed groups after 12 weeks (HFD group) or after an additional 12 weeks in standard (HFD NE) or environmental enrichment (HFD EE) conditions (Figure 3E). Our analysis revealed no significant differences in the diversity (Figure 3F) and richness (Figure 3G) among the HFD groups. This indicates that additional time on the HFD (Appendix A) or housing in an EE (Appendix A) did not notably impact the overall alpha bacterial diversity of the microbiota. However, beta diversity analysis indicated that the microbiota of mice fed a HFD for 12 weeks (HFD) resembled that of HFD EE mice (Figure 3H and Appendix A), indicating that the EE prevented further changes to the microbiota caused by an additional 12 weeks of HFD feeding (Appendix A). Following this, we identified a few differences in the linear discrimination analysis (LDA) comparing the HFD and HFD EE mice (Appendix A). Nevertheless, distinct bacterial taxa enriched in each group were observed when comparing the three HFD groups (Figure 3I). The EE specifically enriched the family Dehalobacteriacea, the genera Rikenella, Syntrophoccocus, and Dehalobacterium, and the species *S. sucromutans* (Figure 3I) compared to HFD mice housed in standard conditions for 12 (HFD) or 24 weeks (HFD NE). Conversely, the EE on the HFD decreased bacteria such as *Alkaliphilus*, *Fusibacter*, SMB53, *Desulfovibrio* C21_c20, and *Lactobacillus reuteri*, which were increased in HFD NE mice (Figure 3I). Interestingly, an increased abundance of these bacteria has been associated with obesity, T2D, inflammation, and cardiovascular diseases [37,39]. Consistent with these microbial shifts, we observed positive metabolic changes in the HFD EE group. The EE led to weight loss (Figure 4A) and prevented an increase in weekly caloric intake (Figure 4B) in HFD-fed mice. It also reduced fasting glucose levels and improved glucose tolerance and insulin sensitivity compared to other HFD groups (Figure 4C–G). Moreover, the EE was effective in reducing triglyceride levels compared to those observed in HFD mice housed in standard conditions for 12 (HFD) or 24 weeks (HFD NE) (Figure 4H). The EE also prevented the increase in serum cholesterol (Figure 4I) and HDL (Figure 4J) levels that occurred with an additional 12 weeks on a HFD (HFD NE group). No significant differences were observed in LDL levels between HFD-fed mice (Figure 4K). Additionally, we found that the EE inhibited the recruitment of infiltrating cells (Figure 4L) and infiltrated areas (Figure 4M) in adipose tissue, maintaining levels observed after only 12 weeks on a HFD. These results indicate that the EE prevents the progression of metabolic alterations due to an additional 12 weeks on a HFD and reduces bacterial taxa associated with metabolic syndrome.

### 2.3. Environmental Enrichment Increases the Levels of Muc2 in the Colon of Mice Fed with a High-Fat Diet

Goblet cells secrete mucus, forming a protective barrier in the intestines that prevents the entry of pathogens and microbial constituents, such as LPS, into circulation, thus preventing systemic inflammation. Dysbiosis of the gut microbiota can increase intestinal permeability [40,41]. Given our current findings that the EE prevents further alterations in the microbiota composition caused by HFD feeding, reduces bacterial taxa associated with metabolic syndrome, and mitigates obesity-associated inflammation in adipose tissue, we examined serum LPS levels as an indicator of gut barrier integrity. Remarkably, the EE condition decreased circulating LPS levels compared to the HFD and HFD NE groups (Figure 5A).

We then investigated whether the EE could reduce LPS levels by preserving the gut epithelial barrier. We found that the EE significantly increased the number of Alcian-blue/PAS-stained goblet cells in mice fed with the HFD (HFD EE) compared to the HFD and HFD NE groups (Figure 5B,C). Similarly, levels in the intestine of the inflammatory marker IL-1β were significantly lower in HFD EE mice compared to the HFD and HFD NE groups (Figure 5D). This reduction aligns with the decreased inflammation observed in the adipose tissue of HFD EE mice (Figure 2F–H). Altogether, these results suggest that the EE’s anti-inflammatory effect in HFD-fed mice is associated with the restoration of goblet cell numbers. This is further supported by the levels of Muc2, a key component of the mucosal barrier [42]. We observed that the EE significantly increased Muc2 levels in mice fed a HFD compared to those housed in standard conditions and fed a HFD for 12 (HFD) or 24 (HFD NE) weeks, respectively (Figure 5E). These data indicate that the endotoxemia and intestinal inflammation resulting from HFD feeding (Appendix A) are reduced by the EE in mice fed a HFD.

### 2.4. Environmental Enrichment Modulates Gut Microbiota Dysbiosis and Increases Specific Intestinal Taxa

To better understand how the EE influences the intestinal microbiota in metabolically compromised mice (Figure 3D), we conducted a beta diversity analysis and found a significant separation between the HFD EE and HFD NE groups (ANOSIM R = 0.6111, *p* = 0.003) (Figure 6A). This indicates that the EE induces significant changes in the intestinal microbiota composition in HFD-fed mice. Within the EE condition, we discovered a reduction in 13 taxa and an increase in 11 taxa (Figure 6B). Specifically, the EE (HFD EE group) led to a reduction in o_SM1D11, Clostridiaceae, Peptococcaceae, Veillonellaceae (Clostridiaceae, Peptococcaceae, and Veillonellaceae belong to the order Clostridiales), *Alkaliphilus*, Clostridiisalibacter, *SMB53* (*Alkaliphilus*, *Clostridiisalibacter*, and *SMB53* belong to the family Clostridiaceae), Acidaminobacteraceae, *Fusibacter*, *Tepidimicrobium*, *Allobaculum*, *Lactobacillus reuteri*, and *Desulfovibrio* C21_c20. Meanwhile, it increased class Sva0725, order Sva0725, Bacteroidaceae, Prevotellaceae, Dehalobacteriaceae (Bacteroidaceae and Prevotellaceae belong to the order Bacteroidales), *Bacteroides*, *Prevotella*, *Butyricimonas*, *Dehalobacterium*, *Syntrophococcus*, and *Syntrophococcus sucrumutans*. We then examined the correlation between the relative abundance of these taxa and metabolic parameters using the Spearman correlation coefficient. Interestingly, we found a negative correlation between body weight gain (BWG) and insulin resistance (IR) with several of the taxa enriched in the HFD EE group (*Syntrophococcus sucromutans*, class Sva0725, order Sva0725, Prevotellaceae, *Prevotella*, Dehalobacteriaceae, *Syntrophococcus*, Dehalobacteriaceae, and *Dehalobacterium*; *Butyricimonas* with BWG only) (Figure 6B and Appendix A). Conversely, we observed a positive correlation between BWG and some taxa enriched in the HFD NE group, such as Acidaminobacteraceae, *Fusibacter*, and *Tepidimicrobium* (Figure 6B and Appendix A). Altogether, these findings indicate that EE restores metabolic imbalance in diet-induced obesity by regulating gut microbiota dysbiosis and increasing specific intestinal taxa associated with a healthy state.

## 3. Discussion

The consumption of a high-fat diet has long been recognized as a significant contributing factor to the development of chronic degenerative diseases, encompassing conditions such as obesity, T2D, cardiovascular diseases, metabolic syndrome, and cancer [3]. In animal models designed to induce obesity through dietary means, the manifestation of these disorders is characterized by a range of physiological changes. These include an increase in body weight, the onset of T2D marked by disruptions in glucose metabolism, specifically glucose intolerance and insulin resistance, the emergence of cardiovascular issues typified by elevated serum levels of cholesterol, triglycerides, and LDL cholesterol, alongside a reduction in HDL cholesterol levels. Metabolic syndrome, a complex interplay of obesity, T2D, and lipid profile abnormalities, co-occurs with these ailments in the human population as well [43]. Moreover, there is a growing body of evidence linking high-fat diets to chronic degenerative diseases, including cancer, through the presence of meta-inflammation, a persistent, low-grade inflammatory process in critical metabolic organs, including the pancreas, skeletal muscle, adipose tissue, the liver, and brain, generating insulin and leptin resistance and alterations in glucose and lipid metabolism [3,4,6].

Within the scope of this study, employing a murine model of diet-induced obesity, we observed that after 12 weeks of high-fat diet consumption, several critical hallmarks of this diet-induced obesity paradigm were evident. These include a substantial increase in body weight, glucose intolerance, insulin resistance, and elevated serum cholesterol and LDL cholesterol levels. These findings validate our obesity–T2D model and corroborate previous studies [3,26].

Under normal physiological circumstances, adipocytes maintain a consistent size and secrete adiponectin, which is an integral component of an anti-inflammatory adipose tissue environment. However, in obesity, meta-inflammation emerges, predominantly within adipose tissue. In this state, adipocytes and macrophages become potent sources of pro-inflammatory molecules, including IL-1β, IL-6, TNF, CCL2, and CCL3. This, in turn, triggers the recruitment of more immune cells, forming characteristic crown-like structures around adipocytes. These structures are organized in clusters of immune cells that encircle and interact with adipocytes that have died due to hypoxia induced by hypertrophy and hyperplasia [44,45]. Our observations in this study align with these patterns, as we noted an increase in adipocyte size within epididymal white adipose tissue. This was accompanied by a notable rise in the proportion and number of infiltrating immune cells, as shown in representative images that capture the formation of crown-like structures. These physiological changes correlate with an overall increase in the weight of epididymal white adipose tissue (HFD: 1.90 g ± 0.209165 g vs. ND 0.475 g ± 0.151544, ** *p* < 0.01; n = 4). The liver plays a vital role in cholesterol and lipid metabolism, although it is not the exclusive organ responsible for these processes. Consumption of a high-fat diet has been shown to have detrimental effects on the liver, leading to a condition known as non-alcoholic fatty liver disease (steatosis). This condition is characterized by the accumulation of fat droplets within liver cells and, if left untreated, can progress to more severe conditions like cirrhosis or liver cancer [46,47,48]. In our study, mice subjected to a high-fat diet for 12 weeks displayed clear signs of fatty liver development, which further validated our diet-induced obesity model.

Recent studies have shed light on the significant role of intestinal microbiota in the development of obesity and T2D [46,47,48]. This connection has been established through various studies, with germ-free mice, i.e., those bred without detectable microorganisms in their gut, serving as a key model. Notably, germ-free mice do not develop obesity when exposed to a high-fat diet. However, when transplanted with fecal material from obese mice, these germ-free mice gain weight, experience increased white adipose tissue, and exhibit disruptions in glucose metabolism [16]. These results point out the importance of the gut microbiota for metabolic balance.

Previously, it has been shown that consumption of a high-fat diet alters the composition of the intestinal microbiota [3,15,39]. Through an analysis of alpha diversity, we observed a decrease in both richness and diversity. Further, beta diversity analysis (unweighted UniFrac PCoA) showed distinct separations between the regular chow and high-fat diet groups, with an R^2^ value of 0.962962 (*p* = 0.002), denoting differential microbial populations. Then, we were particularly interested in identifying taxonomic groups that exhibited differential abundances, which we achieved with the linear discriminant analysis effect size (LDA effect size) algorithm. Within the HFD group, we noted a decrease in 78 taxa, with 24 taxa exhibiting increased abundance. These changes were observed at various taxonomic levels. Some of these taxa correspond with previous findings regarding diet-induced obesity. Notably, a decrease in *Akkermansia muciniphila*, a microorganism associated with a homeostatic state, was observed. Additionally, in the HFD group, we observed increased abundance in taxa such as *Alkaliphilus*, *Fusibacter*, *SMB53*, *Desulfovibrio C21_c20*, and *Lactobacillus reuteri*, which have been previously linked to the development of obesity, T2D, inflammation, and cardiovascular complications [37,49]. It is noteworthy that we did not detect significant differences in the relative abundance of HFD intake-associated phyla such as Firmicutes (regular chow = 0.4696609 and high-fat diet = 0.5066529) and Bacteroidetes (regular chow = 0.39257109 and high-fat diet = 0.40043305). However, the impact of specific taxa on host physiology is highly context-dependent (e.g., strain and vivarium conditions). In fact, identifying the key pathogenic microbiota and establishing a causal (rather than associative) relationship between specific gut microbiota and a given physiological or disease condition remains a significant challenge. In the present study, after the initial 12-week period of consuming either a high-fat or regular diet, the mice were transferred to a regular or environmental enrichment housing condition, where they continued their respective dietary regimens for an additional 12 weeks. Our study replicated the beneficial effects of a previously reported environmental enrichment condition [26]. These include reduced body weight gain, improved glucose tolerance, enhanced insulin sensitivity, decreased inflammation in epididymal white adipose tissue, and reduced levels of pro-inflammatory cytokines such as IL-1β, IL-6, and TNF in the adipose tissue. EE also reduces both the percentage and number of immune cells surrounding adipocytes, effectively reducing adipocyte hypertrophy and hyperplasia, as well as mitigating fatty liver. Notably, unlike the findings in the work by Díaz de León-Guerrero et al. (2022) [26], our mice did not fully recover their body weight in the EE group, and it remained lower compared to the NE group throughout the experiment. Another significant difference was the reduction in serum cholesterol levels. These differences from the work by Díaz de León-Guerrero et al. (2022) [26] could be attributed to the fecal microbiota homogenization applied in the present study.

It is well-documented that high-fat diet consumption induces inflammation in the colon of animal models [3,50,51]. Therefore, we were interested in exploring the impact of an EE on the colon of mice subjected to a high-fat diet. As expected, animals in the high-fat diet group (HFD NE group) exhibited a structure characterized by enlarged crypts. Strikingly, exposure to EE recovered the structure of the proximal colon in mice maintained in HFD feeding. This observation correlates with a reduction in the colon inflammatory process resulting from HFD feeding, as evidenced by decreased levels of the pro-inflammatory cytokine IL-1β in the HFD EE group. These results align with our previous findings, which demonstrated the anti-inflammatory effects of EE in both the colon of a mouse colitis model and in the adipose tissue of obese mice fed a HFD [26].

It is well-established that a HFD reduces the number of goblet cells responsible for mucous production [52]. Interestingly, we discovered that the number of goblet cells increased with EE exposure, even in mice fed a HFD. Noteworthy, this increase in goblet cell numbers correlates with reduced levels of LPS in the serum of mice fed a HFD and maintained in an EE. Elevated LPS levels in the bloodstream have been associated with increased inflammation in the intestine, mainly due to the disruption of the intestinal barrier [40,41]. This barrier ordinarily houses the gut microbiota but becomes compromised with factors such as HFD consumption, leading to endotoxemia [53]. This disruption reduces tight junction proteins like occludin, claudin-1, -3, -4, and -7, alongside an increase in claudin-2 [54,55]. These alterations facilitate the passage of immune material and bacteria, including LPS, from the gut microbiota into the bloodstream. This process leads to inflammation in several regions of the intestine and other organs, and it has been associated with insulin resistance [11,53,54,55,56,57]. Furthermore, disruptions in intestinal homeostasis result in dysbiosis [57]. Even though we did not see differences in the tight junction protein levels in the colon of the HFD EE mice, we found increased levels of the mucin Muc2. This is particularly interesting since depletion [58] or reduced production of mucosal-barrier-related molecules, such as Muc2 and goblet cells, has been associated with systemic inflammatory responses and the appearance of several diseases like colorectal cancer, Crohn’s disease, ulcerative colitis, obesity, and T2D [42,59]. Our study suggests that an EE enhances intestinal barrier protection by increasing goblet cell numbers, leading to elevated mucus production.

Interestingly, mice fed regular chow (ND) and exposed to either environment (NE or EE) did not display significant differences in bacterial richness and diversity in the alpha diversity analysis. These mice also did not exhibit significant differences in the beta diversity analysis (unweighted UniFrac PCoA), with an R^2^ value of 0.129629, consistent with previous observations [27]. However, we did identify differentially abundant bacterial taxa between these two experimental groups (ND NE and ND EE). In the ND NE group, nine taxa showed differential abundance, while the ND EE group exhibited two taxa with such variations. Notably, *Bacteroides* were increased in the ND EE group. This genus has been associated with a healthy state [60].

The beneficial effects of an EE in preventing and reversing obesity and T2D in animal models have been reported [25,26], and there is also evidence of its impact on reducing inflammation in the colon of a mouse colitis model and altering the composition of intestinal microbiota in murine Parkinson’s disease model [27]. Like the findings in the Parkinson’s study, we did not observe significant differences in bacterial richness and diversity in the intestinal microbiota between the HFD NE and HFD EE groups. However, a notable distinction arose in the beta diversity analysis (unweighted UniFrac PCoA), which revealed an R^2^ value of 0.661111 (*p* = 0.003), indicating a clear separation between the two groups in this analysis. This indicates that, with the primary variant being the environment, the main factor driving the microbiota changes was the diet (R^2^ = 0.962962, *p* = 0.002), followed by the environment. Subsequently, we conducted an analysis using LEfSe, which identified the differentially abundant taxa between the HFD NE and HFD EE groups. Of particular interest was that the bacteria over-abundant in EE negatively correlated with body weight gain and insulin resistance, whereas those over-abundant in NE positively correlated with these parameters. This suggests that the intestinal microbiota may be a critical factor in mediating these effects in animals exposed to an EE. Interestingly, this finding correlates with a reduction in intestinal permeability (reduced LPS levels) observed in obese mice that were fed an HFD and maintained in an EE. This observation may also provide insight into the beneficial effects of EE in preventing obesity in animal models [25]. Similarly, previous studies reported the positive effect of an EE in modulating inflammation in the intestine and modulating gut bacteria in Parkinson’s disease [27] and colon cancer [61] mouse models.

In the HFD EE group, 13 taxa decreased, and 11 taxa increased in abundance, spanning various taxonomic levels. Among the taxa that decreased in HFD EE but increased in HFD NE, we found *Alkaliphilus*, *Clostridiisalibacter*, *SMB53*, *Fusibacter*, and *Allobaculum*. Interestingly, these bacteria increased their abundance with a HFD and were associated with the development of T2D, hypercholesterolemia, and metabolic syndrome [62,63]. Importantly, Clostridiales was reported as over-abundant in obesity with metabolic syndrome compared to normal-weight children and was being suggested as a biomarker of obesity [64]. *Lactobacillus reuteri* (s_reuteri) was also increased in the HFD NE. The L10 strain isolated from obese mice derives from *Lactobacillus reuteri* [38]. This suggests that under our experimental conditions, the HFD NE mice may have an increased abundance of L10 and not the PTA6475 strain, which has beneficial effects in a murine model of autism [23]. The increased abundance of *Desulfovibrio C21_c20* observed in the HFD group was notably reduced following exposure to EE. Some species of *Desulfovibrio* have been linked to the development of obesity, T2D, and an inflammatory process [65,66]. This reinforces our hypothesis about the role of an EE in modulating the composition of intestinal microbiota and suggests that some of the beneficial effects of this environment could be attributed to the constitution of the intestinal microbiota.

Taxa whose abundance was increased in the HFD EE group, such as *Syntrophococcus*, *S. sucromutans*, class Sva0725, order Sva0725, Prevotellaceae, *Prevotella*, Dehalobacteriaceae, *Dehalobacterium*, and *Butyricimonas* negatively correlated with body weight gain and insulin resistance. Interestingly, the administration of probiotics such as *Lactobacillus* to treat cirrhosis also increases the abundance of *Syntrophococcus sucromutans* and *Prevotella*, reducing cirrhosis [67]. So far, there is no available information regarding any beneficial effects of *Syntrophococcus sucromutans* on obesity or hypercholesterolemia, but intriguingly, it degrades lignin, cholesterol, and long-chain fatty acids into short-chain fatty acids. These short-chain fatty acids serve as an energy source for intestinal epithelial cells and other bacteria, a phenomenon observed in bacterial cultures supplemented with fatty acids and in the rumen of cows [68,69,70]. Furthermore, its abundance has been positively correlated with antioxidant, anti-inflammatory markers, and energy expenditure [71]. On the other hand, the administration of soluble corn fiber and fructooligosaccharides increases the abundance of *Dehalobacterium*, which leads to a reduction in body weight gain and insulin resistance in a diet-induced obesity murine model [72]. Reduced abundance of this bacterium has been associated with increased inflammation [73].

In female ovariectomized mice, inoculation with *Prevotella histicola* increases tight junction proteins such as ZO-1, occludin, claudin-1, and Muc2 in the colon [21]. It also reduces levels of inflammatory factors like MCP-1, IL-6, IL-8, and TNF and increases BDNF levels in the hippocampus [21]. Additionally, it is noteworthy that administering *Butyricimonas virosa* effectively prevents body weight gain and mitigates metabolic disorders in a murine model of diet-induced obesity [74]. These findings underscore the significance of specific microbial interventions in modulating gut health. Further studies are required to ascertain whether specific bacteria, such as *Syntrophococcus sucrumutans*, *Dehalobacterium*, *Prevotella*, and *Butyricimonas*, can replicate the beneficial effects observed in the EE group within this study.

Moreover, the taxa whose abundance increased in the HFD NE group but decreased in the HFD EE group, including *Tepidimicrobium*, Acidaminobacteraceae, and *Fusibacter*, exhibit a positive correlation with body weight gain. Notably, these bacteria have been identified in individuals at risk of metabolic syndrome [75].

The role of an EE in modulating the gut microbiota likely involves decreasing the abundance of acetate-producing bacteria and increasing those that produce bile acids and other short-chain fatty acids, such as butyrate. In this sense, treatment of rats with sodium butyrate reduces plasma glucose levels, insulin resistance, fat accumulation in WAT, and liver steatosis, showing similar effects to an EE [76].

In many developing countries, lifestyles often fail to support a healthy diet. This study confirms the therapeutic potential of an EE in mitigating energy imbalance, even when organisms maintain a HFD. The metabolic protection afforded by an EE is closely tied to its anti-inflammatory properties, which not only prevent further alterations in the gut microbiota but also foster the proliferation of specific bacterial populations associated with maintaining homeostasis. This is particularly interesting since recent work has established intestinal barrier damage with intestinal and systemic inflammation in obese T2D humans fed with a lipid diet [77]. Therefore, an EE might constitute a preventive/therapeutic strategy to modulate intestinal barrier function in metabolic diseases.

Additionally, our study provides evidence supporting the development of a probiotic based on the specific bacterial taxa enriched by an EE (*Syntrophococcus sucrumutans*, *Dehalobacterium*, *Prevotella*, and *Butyricimonas*) to reduce inflammation in obesity-related metabolic complications, although further studies are required to achieve this goal.

Altogether, here we demonstrate that EE intervention prevents the progression of HFD-induced dysbiosis, enhances intestinal barrier integrity, increases the mucin-producing goblet cell population, and upregulates Muc2 expression in the colon. These findings correlate with reduced systemic LPS levels and attenuated colon inflammation, resulting in normalized glucose metabolism, diminished adipose tissue inflammation, reduced liver steatosis, improved lipid profiles, and a significant reduction in body weight gain despite the mice’s continued HFD consumption. Our findings highlight the effects of an EE as an immunometabolic protector. Likewise, the decrease in body weight suggests EE neuromodulatory effects. Previously, we reported that an EE promotes WAT browning and increases anorexigenic markers (POMC and CART) in the hypothalamus [25,26]. Both neurohormones are implicated in feeding behavior and energy expenditure. Ongoing experiments in our group aim to determine whether these anorexigenic signals are part of the beneficial EE effect on obese mice to regulate body weight.

Most research has focused on the effects of a complete EE in various pathologies. However, some studies have attempted to elucidate the contributions of different components of an EE. In certain models, exercise has been shown to account for most of the beneficial effects of an EE [78,79]. Conversely, other studies have indicated that the combination of different components, e.g., physical, social, or cognitive stimulation, provides a greater effect than each component alone [80,81]. Consequently, exercise alone cannot replicate the full effects of a complete EE on reducing adiposity or modifying gene expression in the hypothalamus and various fat depots [77]. Overall, these findings suggest that the specific contributions of each component of an EE to the observed beneficial effects under our experimental conditions require further characterization.

Translating findings from murine models to human treatments presents challenges; however, several studies have shown beneficial EE outcomes in clinical contexts, spanning neurodevelopmental disorders, age-related cognitive decline, and neurodegenerative diseases [31,32,33,34]. This study shows the anti-inflammatory and therapeutic properties of our EE protocol. Consequently, we advocate for the incorporation of sensory therapy rooms into clinical practice, integrating elements akin to EE components, as previously described [31,32,33,34]. These rooms would encompass physical, cognitive, social, and somatosensory stimulation, offering supplementary interventions to current treatment modalities aimed at addressing metabolic dysregulations in obese/diabetic patients. Such interventions could potentially prevent the onset of other pathologies, including cardiovascular and neurodegenerative diseases, cancer, and T2D.

Finally, our research has provided valuable insights into the intricate interplay among diet, environmental enrichment, and microbiota in the context of metabolic health in a murine model. These findings highlight potential mechanisms underlying the beneficial effects of an EE in countering obesity and insulin resistance. Further investigations into the roles of specific bacterial strains and their functionalities in mediating these effects could unveil new therapeutic opportunities for metabolic diseases. Moreover, a comprehensive exploration of microbiota changes in response to diet and the EE lays a solid foundation for future research in this field.

## 4. Materials and Methods

### 4.1. Animals

C57BL/6N male mice at 3–4 weeks old were housed in maternity boxes (21 cm width × 29 cm long × 16 cm height) with 15 mice per cage. They were provided with a regular chow diet (ND, 60 mice) (2018S Envigo Teklad Global with 18 kcal% fat, 58 kcal% carbohydrate, and 24 kcal% protein). Fecal samples from different boxes were combined into a mix, which was then redistributed into each of the boxes to take advantage of the coprophagic characteristic of mice to induce gut microbiota homogenization. Once the mice reached 7–8 weeks of age, they were randomly allocated into two dietary groups: standard housing (NE) and fed either ND (30 mice) or a high-fat diet (HFD, 30 mice) (research diets, D12492 with 60 kcal% fat, 20 kcal% carbohydrate, and 20 kcal% protein). After 12 weeks, five mice from each group were euthanized using CO_2_ inhalation, and their blood and tissues were collected for subsequent analyses. The remaining mice were again randomly assigned, with some kept in the NE, while others were transferred to environmental enrichment (EE) while continuing the same diet. This led to the formation of four experimental groups: ND NE (regular chow diet standard environment, 10 mice); ND EE (regular chow diet environmental enrichment, 15 mice); HFD NE (high-fat diet standard environment, 10 mice); HFD EE (high-fat diet environmental enrichment, 15 mice). After the switch in housing conditions, mice were further maintained for an additional 12 weeks. The EE housing conditions comprised spacious cages (32 cm width × 88 cm long × 47.6 cm height per cage) with 15 mice per cage. These cages were enriched with tunnels, plastic balls, Legos, and wooden structures. The environmental setup was changed weekly to maintain novelty. Mice were subjected to a regular 12 h light and 12 h dark cycle and provided with the respective diet and water ad libitum. Weekly, mice and food leftovers were weighed to record individual weight gain, food consumption per mouse, and weekly water consumption per mouse. Food intake and water consumption per cage were measured, and averages per mouse per week were calculated. Metabolic tests were conducted at experimental weeks 0, 11, and 12 for the ND and HFD groups before the introduction of EE conditions and at weeks 23 and 24 following the transition to an EE. The animals were sacrificed by CO_2_ inhalation after 13 (ND and HFD groups) or 25 (ND NE, ND EE, HFD NE, and HFD EE groups) weeks in the experimental settings. Blood and tissues were collected for further analysis. All animal experiments described in this study were approved by the Institutional Bioethical Committee, protocol number 300.

### 4.2. Basal Glucose Levels, Glucose Tolerance, and Insulin Resistance Test

Basal glucose levels were measured in randomly selected mice at experimental weeks 0, 11, 12, 23, and 24 following a 6 h fast (from 8:00 to 14:00 h). Blood glucose levels were determined using a glucometer (ACCU-CHECK Active^®^ from Roche, Basel, Switzerland), as described below.

The glucose tolerance test (GTT) and insulin resistance test (ITT) were conducted during experimental weeks 11–12 and 23–24. Randomly selected fasted mice received intraperitoneal injections of glucose (1.8 mg per gram of mouse weight) or insulin (1 U/kg; Humulin R, Eli Lilly, Indianapolis, IN, USA) diluted in sterile 1X PBS. Blood glucose values were recorded at 15, 30, 60, and 120 min following the administration of glucose or insulin.

Area under curve

The area under the curve (AUC) for both the GTT and ITT was calculated using Tai’s formula [82]:$$Area=\frac{1}{2}\sum_{i=1}^{n}X_{i}-X_{\left(i-1\right)}\left(Y_{\left(i-1\right)}+Y_{i}\right),$$
where *X* is the time (0, 15, 30, 60, and 120 min), and *Y* is the glucose (mg/dL) level at each time point.

### 4.3. Serum Harvesting

Blood samples obtained through cardiac puncture were allowed to incubate at 4 °C for 2 h to promote clot formation. Subsequently, they were centrifuged at 1200 rpm for 10 min. The serum was carefully collected and stored at −70 °C until further use.

### 4.4. Lipids Determination

Lipids levels were determined in serum (100 μL). Total cholesterol and triglyceride (TG) levels were quantified using the Spinreact kit (Technology for Life, Barcelona, España) and the Triglycerides Roche-Cobas C111 analyzer kit (Roche Diagnostics, Indianapolis, IN, USA) following the manufacturer’s instructions. High-density lipoprotein cholesterol (HDL-C) was measured using the HDL-C plus 3rd generation Roche-Cobas C111 kit (Roche Diagnostics, Indianapolis, IN, USA) following the manufacturer’s instructions. Low-density lipoprotein (LDL) was calculated using Friedewald’s formula [83].

### 4.5. Mice Perfusion

Mice were intraperitoneally anesthetized with ketamine (10 mg/Kg of body weight) and 10% xylazine (80 mg/Kg of body weight). Subsequently, by ascending the aorta, a 0.9% sodium chloride solution was perfused for 15 to 20 min, depending on whether the mouse was lean or obese, followed by 4% paraformaldehyde. Both solutions were administered at a flow rate of 3.5 to 5 mL/min. The liver, epididymal white adipose tissue, and proximal colon were then harvested, fixed in 4% paraformaldehyde, and stored at 4 °C for subsequent histological analysis.

### 4.6. Tissue Analysis

For histological analysis, tissues were embedded in paraffin, and 5 μm histological sections were prepared. These sections were then stained with hematoxylin-eosin (for adipose tissue, liver, and proximal colon) or PAS/Alcian blue (for the proximal colon).

Microphotographs were taken using a Zeiss Axioskop Epifluorescence microscope (Zeiss, Oberkochen, Alemania) with 10× and 20× objectives, equipped with a Moticam 2 camera (Motic^®^, Hong Kong, China) at the National Laboratory for Advanced Microscopy (LNMA), UNAM. The percentage of infiltrated cells in the adipose tissue was determined using ImageJ software 53t. Images of equal size (1280 × 960 pixels) were used to set a scale, and the total area was calculated. Subsequently, the image was converted to 8-bit, and a threshold was applied to estimate the percentage, following the guidelines provided in the manual (http://rsbweb.nih.gov/ij/) accessed on 30 December 2021. The count of infiltrated immune cells was determined by counting the number of cells in 10 fields (20×) from three slides of each individual mouse, with a total of three mice per group.

Representative images of the colon were captured using the Zeiss Axioskop Epifluorescence microscope (20×) and a Moticam 2 camera. For goblet cell quantification, an optical microscope (Nikon Eclipse TS100) (Nikon Instruments Inc., Tokyo, Japan) with a 40× objective and a Sony DSC-HX1 camera (SONY, Tokyo, Japan) was used. Goblet cells were quantified per area by calculating the infiltration in 20 fields (40×) from three slides of each individual mouse, with a total of three mice per group. The photographs were analyzed to determine the area corresponding to goblet cells using FIJI software 2.3.0/1.53f. The procedure involved verifying that the images were of the same size (1280 × 960 pixels), setting the scale, measuring the total area from the photograph, converting the picture to 8-bit, and finally applying a threshold to measure the infiltrated area, following the instructions provided in the manual (http://rsbweb.nih.gov/ij/) accessed on 30 September 2020 and 14 December 2021.

### 4.7. Total Protein Extraction and Western Blot Assay

Total protein was extracted from epididymal adipose tissue and the proximal colon and then stored at −70 °C for further use. Protein extracts were prepared in a lysis buffer containing 20 mM Tris (pH 7.4), 137 mM NaCl, 25 mM β-glycerophosphate (pH 7.4), 2 mM PPiNa, 2 mM EDTA (pH 7.4), 1% Triton X-100, and 10% glycerol. The complete protease inhibitors cocktail (Roche) and phosphatase inhibitors (200 mM Na_3_VO_4_, 0.1 mM DTT, and 1mM of PMSF) were also added, and the samples were subjected to sonication (10 pulses of one second each for three to four times at 90% amplitude). The lysates were then incubated for 15 min on ice and subsequently centrifuged for 10 min at 14,000 rpm, and the supernatant was collected. The protein concentration was determined using the Bradford method with a BSA standard curve.

Proteins (30 µg) were separated on polyacrylamide gels and transferred to a nitrocellulose membrane (Hybond-ECL, GE Healthcare Life Sciences, Marlborough, MA, USA) at 45 mAmp for 90 min. The membranes were blocked with 5% skim milk in TBS-T (20 mM Tris pH 7.5, 150 mM NaCl, and 0.1% Tween) for one hour at room temperature. Then, the primary antibody was incubated overnight at 4 °C. After three washes with TBS-T, membranes were incubated for one hour at room temperature with the secondary antibody. After an additional three washes with TBS-T, the immunocomplexes were visualized through chemiluminescence using an LI-COR C-DiGit^®^ Blot Scanner (Biosciences instrument, Groot-Ammers, The Netherlands). Densitometry analysis was performed using Image Studio Software version 5.2.5 (LI-COR Biosciences). Primary antibodies used Muc2 (H-300, 1:500 dilution; Santa Cruz Biotechnology, Dallas, TX, USA, sc-15334) and actin (1:10,000 dilution; Cell Signaling Technology, Danvers, MA, USA, 3700). Secondary antibodies used anti-rabbit IgG (H+L) peroxidase-conjugated horseradish (1:5000 dilution, Life Technologies, Carlsbad, CA, USA, G21234) and anti-mouse IgG (H + L) peroxidase-conjugated horseradish (1:15,000 dilution, Santa Cruz Biotechnology, sc358914)

### 4.8. ELISA

The levels of IL-6, IL-1β, IL-10, and TNF in the adipose tissue and IL-1β in the proximal colon were quantified using ELISA MAXTM Deluxe Set Mouse TNF-α (Cat. # 430904), ELISA MAXTM Deluxe Set Mouse IL-6 (Cat. # 431304), ELISA MAXTM Deluxe Set Mouse IL-10 (Cat. # 431414), and ELISA MAXTM Deluxe Set Mouse IL-1β (Cat. # 432604) kits, following the kit instructions. For all ELISA measurements (IL-6, IL-1β, IL-10, and TNF), the procedure involved the following steps: 100 μL of diluted capture antibody solution were added to each well, followed by overnight incubation between 2 °C and 8 °C. The plate was washed four times, and then it was blocked with 200 μL of 1X assay diluent A in each well. This was incubated at room temperature for 1 h with shaking at approximately 500 rpm. The plate was washed again four times, and 100 μL of diluted standards and samples (proteins extracted from the WAT, as mentioned previously) were added to the wells. This was followed by incubation at room temperature for 2 h with shaking. The plate was washed four times, and 100 μL of diluted detection antibody solution were added to each well, followed by incubation at room temperature for 1 h with shaking. After this step, the plate was washed four times again. Then, 100 μL of diluted avidin-HRP solution were added to each well, followed by incubation at room temperature for 30 min with shaking. The plate was washed five times, with each wash involving soaking for 30 s to 1 min. In total, 100 μL of freshly mixed substrate solution were added to each well, and the plate was incubated in the dark for 20 min for IL-6 and 15 min for IL-1β. Finally, 100 μL of stop solution were added to each well, and the absorbance was determined at 450 nm and 570 nm within 15 min.

### 4.9. LPS Quantification

LPS was determined in serum using the Endotoxin Quantitation Kit (Pierce™ LAL Chromogenic, Rockford, IL, USA) following the manufacturer’s instructions. Briefly, reagents were brought to room temperature before use, and the microplate was pre-equilibrated at 37 °C for 10 min. In total, 50 μL of each standard or sample replicate were dispensed into the microplate well. At time 0, 50 μL of LAL (Limulus Amebocyte Lysate) reagent were added to each well. The plate was covered with a lid and incubated at 37 °C for 10 min. After exactly 10 min, 100 μL of chromogenic substrate solution were added to each well. The plate was again covered with a lid and returned to a heating block to incubate at 37 °C for 6 min. After 16 min in total, 100 μL of the stop reagent (25% acetic acid) were added to each well. Each time reagents were added, the plate was removed from the heating block and gently tapped several times to mix.

Finally, the absorbance at 405–410 nm was measured on a plate reader. A standard curve was prepared by plotting the average blank-corrected absorbance for each standard on the y-axis against the corresponding endotoxin concentration in EU/mL on the x-axis. This standard curve was then used to determine the endotoxin concentration of the samples.

### 4.10. Stool Collection

Stool samples from the mice were collected at various time points: one day before starting HFD administration (experimental week 0), after HFD feeding (experimental week 12), and after exposure to an EE (experimental week 24). The collection tubes contained 300 µL of RNAlater™ (SIGMA-ALDRICH, St. Louis, MO, USA). The collected feces was stored at 4 °C for 24 h and subsequently stored at −70 °C.

### 4.11. Stool Preparation for 16S rRNA Sequencing

For 16S rRNA gene sequencing, six individual samples from each experimental group were used. DNA from the collected stool samples were obtained using the Quick-DNATM Fecal/Soil Microbe Miniprep kit (Cat #D6010, Lot. No. ZRC202269), with all DNA concentrations adjusted to 12.5 ng/µL. The first PCR was performed using Illumina oligonucleotides for the 16S rRNA gene with the forward primer = 5′ (TCG TCG GCA GCG TCA GAT GTG TAT AAG AGA CAG CCT ACG GGN GGC WGC AG) and the 16S amplicon PCR reverse primer = 5′ (GTC TCG TGG GCT CGG AGA TGT GTA TAA GAG ACA GGA CTA CHV GGG TAT CTA ATC C), corresponding to a ~550 bp fragment from the V3–V4 hypervariable regions. Subsequently, all amplicons were purified with Agencourt AMPure XP-PCR purification beads (Beckman Coulter, Brea, CA, USA, Cat. A63882), followed by electrophoresis in a 2% agarose gel to confirm quality DNA. DNA concentration was determined using a Qubit 2.0 fluorometer (Thermo Scientific, Waltham, MA, USA, Q32851). Amplicon sequencing was performed on an Illumina MiSeq platform at the National Institute of Genomic Medicine (INMEGEN) using a 2 × 250 paired-end sequencing format.

### 4.12. Bioinformatic Analysis of the 16S rRNA Profiling Data

Trimmomatic (version 0.36) was used for quality filtering with the following parameters: a sliding window quality of 6nt with an average quality > Q20 Phred Score, removal of sequencing adapters, and ambiguous bases. The resulting reads were joined using fastq-join to obtain complete amplicon sequences of ~550 nt. The QIIME 1.9.1 pipeline was used to cluster amplicon sequences at 97% identity into operational taxonomic units (OTUs). The GreenGenes (version 13_8) database and the UCLUST algorithm were used with closed-reference clustering, allowing for reverse strand matches. Sequences that did not align with the references were not considered for downstream analysis. OTUs with ≤0.005% of the total read abundance were eliminated. Relative abundance was determined by reads and valid taxonomy. The data were stratified into different taxonomic levels (phylum, class, order, family, genus, and species) using database information. QIIME v1.9.1 pipeline scripts were used to calculate indexes for alpha diversity (Shannon, Chao1, observed OTUs, and phylogenetic diversity) and beta diversity (weighted UniFrac, unweighted UniFrac, Bray–Curtis, and Jaccard). The ggplot2 package and R language were used for creating final plots for taxonomy and alpha and beta diversity.

Differentially abundant taxa were identified using the LDA effect size algorithm (LEfSe Galaxy Version 1.0) with a minimum LDA score of 2 and a *p*-value < 0.05 as a cutoff. These analyses were conducted by contrasting between different experimental groups.

Code availability:

The GitHub repository https://github.com/LuiguiGallardo/rubiceli_microbiome_16S contains the code used for 16S rRNA analysis (accessed on 9 June 2024).

### 4.13. Statistical Analysis

Data are presented as means ± standard error or standard deviation. An unpaired *t*-test was used to compare the ND and HFD groups. Two-way ANOVA followed by the Bonferroni post-hoc test was conducted to assess the differences between the ND NE, ND EE, HFD NE, and HFD EE groups. Paired Wilcoxon tests were used to contrast alpha diversity within different groups. Data were also analyzed using PERMANOVA (permutational multivariate analysis of variance), ANOSIM (analysis of similarities), and distance-to-centroid tests. These tests were employed to assess the beta diversity or diversity between groups. *p* < 0.05 was considered statistically significant. 

## Figures and Tables

**Figure 1 ijms-25-06904-f001:**
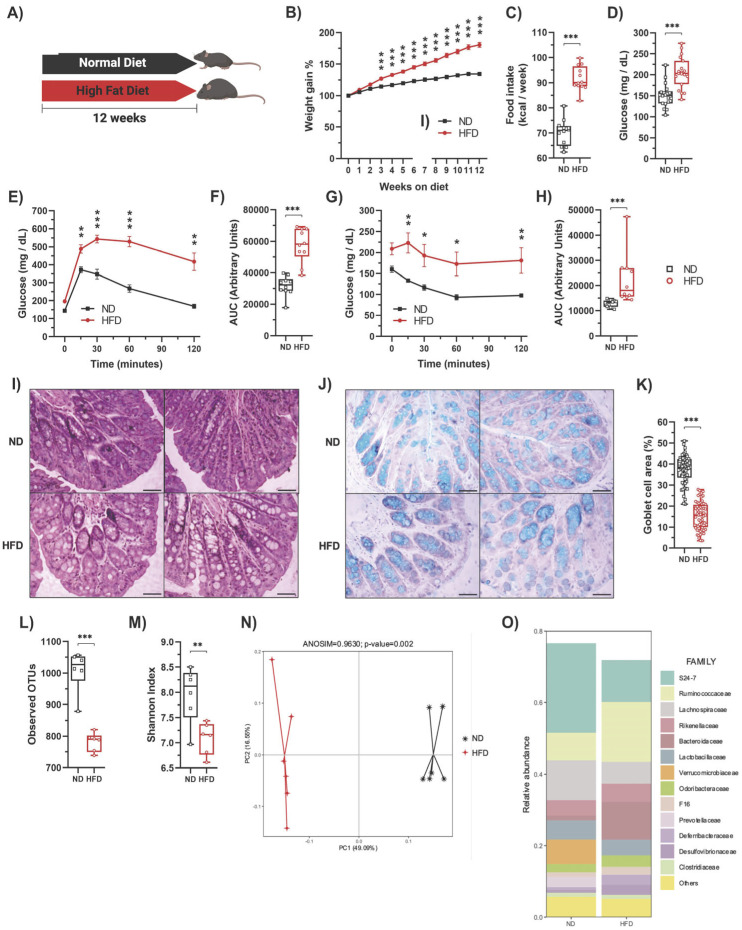
**High-fat diet feeding induces metabolic alterations and decreases the number of goblet cells in mice.** (**A**) C57BL/6N mice were fed either a regular chow diet (ND) (n = 30) or a high-fat diet (HFD) (n = 30) for 12 weeks in standard housing conditions. (**B**) Average weekly weight of mice fed with a ND or HFD. (**C**) Average weekly food intake of mice fed with a ND or HFD. (**D**) At week 12, mice were fasted for 6 h to measure blood glucose levels (n = 20). Then, the glucose tolerance test (GTT) and insulin tolerance test (ITT) were determined. (**E**) GTT (n = 10). (**F**) AUC for the GTT (n = 10). (**G**) ITT (n = 10). (**H**) AUC for the ITT (n = 10). Representative photos of the proximal colon stained with hematoxylin-eosin (**I**) and Alcian blue for acidic mucin in the goblet cells (**J**). Scale bar: 100 μm. (**K**) Percentage of goblet cell area in the proximal colon determined by Alcian blue staining. Alpha diversity evaluation of gut microbiota diversity and richness by measuring the operational taxonomic units (OTUs; (**L**)) and Shannon index (**M**). (**N**) HFD significantly changed the microbiota composition shown by principal component analysis. (**O**) Microbiota composition expressed as the relative abundance of the main families (n = 6). Bars represent the mean ± SEM. * *p* < 0.05, ** *p* < 0.01, and *** *p* < 0.001 vs. ND mice (two-way ANOVA followed by a Bonferroni post-hoc test (**B**,**E**,**G**) or unpaired *t*-test (**C**,**D**,**F**,**H**,**K**,**L**,**M**)).

**Figure 2 ijms-25-06904-f002:**
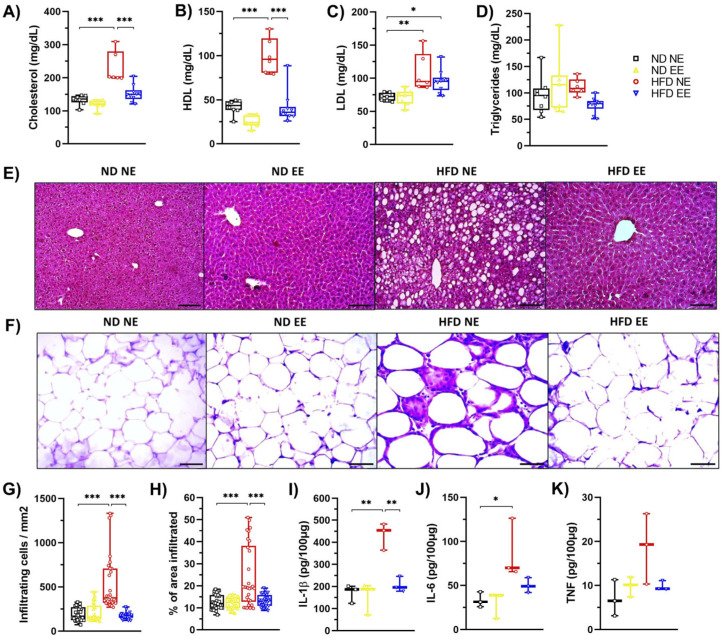
**Environmental enrichment reverses the metabolic syndrome induced by high-fat diet consumption.** C57BL/6N mice in standard housing conditions were fed with a regular chow diet (ND) or high-fat diet (HFD) for 12 weeks and then were separated into standard housing conditions (NE) or environmental enrichment (EE) for an additional 12 weeks. Mice were fed the same diet they had before they were separated into different housing conditions. Experimental groups: ND NE, ND EE, HFD NE, and HFD EE. At week 25, mice were euthanized; tissue, gut microbiota, and serum were collected for further analysis. (**A**) cholesterol, (**B**) HDL, (**C**) LDL, and (**D**) triglyceride serum levels. (**E**) Representative photos of hematoxylin-eosin staining of the liver (n = 3). Scale bars: 100 μm. (**F**) Representative photos of hematoxylin-eosin staining of epididymal white adipose tissue (n = 3). Scale bar: 100 μm. (**G**) Percentage of infiltrating cell count and (**H**) infiltrated area in white adipose tissue. Immune cell infiltration was calculated from 3 mice from each group in 10 different fields (20× magnification). White adipose tissue protein levels of IL-1β (**I**), IL-6 (**J**), and TNF (**K**) determined by ELISA (n = 3). Bars represent the mean ± SEM. * *p* < 0.05, ** *p* < 0.01, and *** *p* < 0.001 vs. ND mice (two-way ANOVA followed by Bonferroni post-hoc test (**A**–**D**,**G**–**K**)).

**Figure 3 ijms-25-06904-f003:**
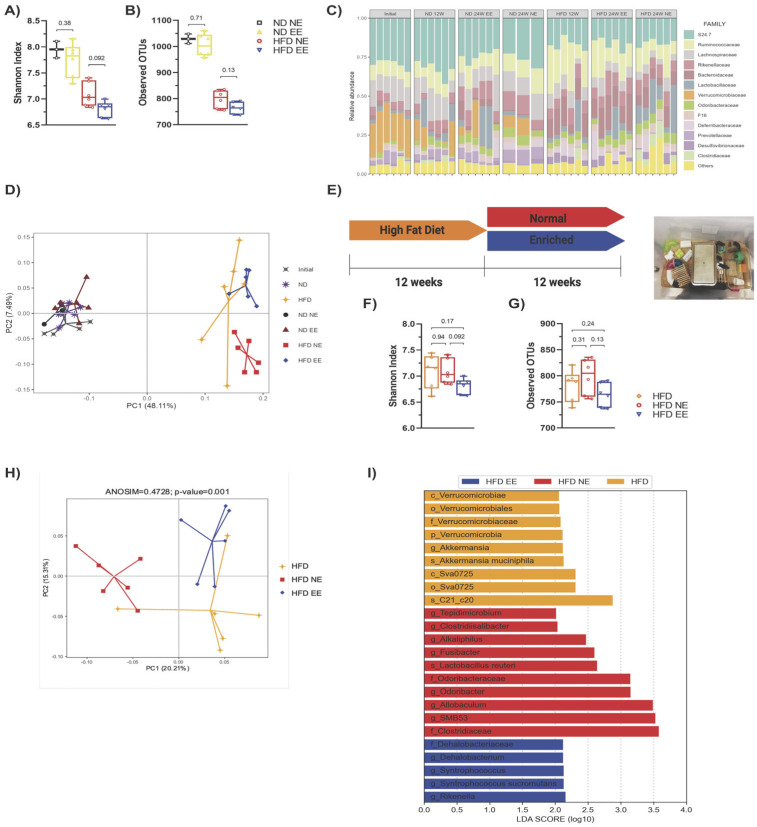
**Environmental enrichment prevents the progression of gut microbiota dysbiosis induced by long-term high-fat diet consumption.** The gut microbiota from the different experimental groups was subjected to high-throughput sequencing targeting the V3 and V4 regions of the 16S rRNA gene (n = 6 from each group). Alpha diversity evaluation of gut microbiota richness and diversity by measuring the Shannon index (**A**) and operational taxonomic units (OTUs; (**B**)). (**C**) Microbiota composition expressed as the relative abundance of the main families. (**D**) Principal component (coordinate) analysis showing the beta diversity clustering of the gut microbiota from mice fed with a regular chow diet (ND) or with a high-fat diet (HFD) and maintained in standard housing conditions (NE) or environmental enrichment (EE) for 12 (ND and HFD) or 24 weeks (ND NE, ND EE, HFD NE, and HFD EE). The gut microbiota at the beginning of the experiment is shown (initial). (**E**) Experimental design. C57BL/6N mice in standard housing conditions were fed a HFD for 12 weeks and then separated into standard housing conditions (normal) or EE for an additional 12 weeks (24 weeks in total). Mice were fed a HFD during the entire experiment. Experimental groups: HFD for 12 weeks in standard housing conditions (HFD); HFD in standard housing conditions maintained for an additional 12 weeks (normal; HFD NE, 24 weeks total); HFD in EE maintained for an additional 12 weeks (enriched; HFD EE, 24 weeks total). Representative photo of an EE cage (right). Alpha diversity evaluation of gut microbiota richness and diversity by measuring the Shannon index (**F**) and operational taxonomic units (OTUs; (**G**)). (**H**) Principal component analysis showing the beta diversity clustering of the gut microbiota from mice fed a HFD for 12 (HFD) or 24 weeks (HFD NE and HFD EE). (**I**) Linear discrimination analysis (LDA) effect size (LEfSe) analysis comparing bacterial phylum, class, order, family, genus, and species between the HFD, HFD NE, and HFD EE groups. Bars represent the mean ± SEM (two-way ANOVA followed by a Bonferroni post-hoc test (**A**,**B**,**F**,**G**)). p_: phylum; c_: class; o_: order; f_: family; g_: genus; s_: species. See Appendix A for the taxonomic ranks of the bacteria.

**Figure 4 ijms-25-06904-f004:**
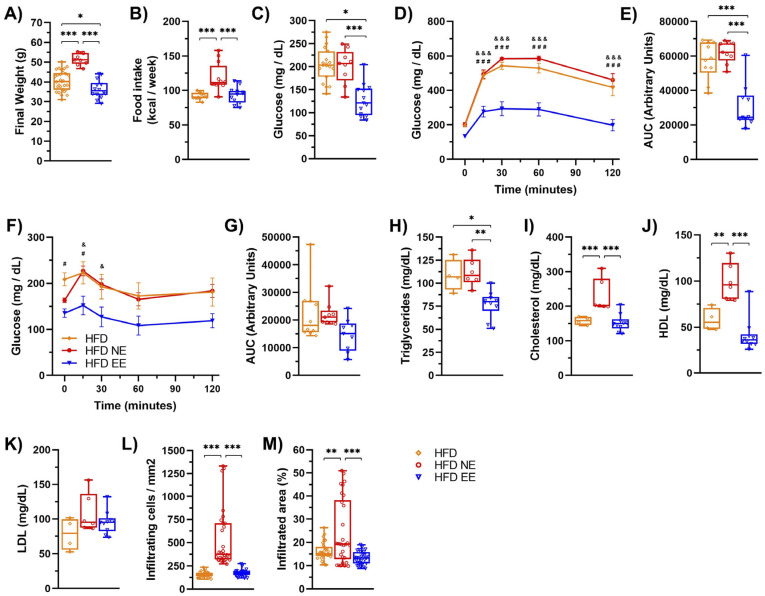
**Environmental enrichment arrests the progression of metabolic damage induced by high-fat diet consumption.** C57BL/6N mice in standard housing conditions were fed a HFD for 12 weeks and then separated into standard housing conditions (NE) or environmental enrichment (EE) for an additional 12 weeks. Mice were fed a HFD during the entire experiment. (**A**) Final weight at week 12 for the HFD group (n = 30) or at week 25 for the HFD NE (n = 10) and HFD EE (n = 15) groups. (**B**) Weekly average of food intake for the HFD, HFD NE, and HFD EE groups. (**C**) After 12 weeks, mice were fasted for 6 h to measure blood glucose levels. Then, the glucose tolerance test (GTT) and insulin tolerance test (ITT) were determined. (**D**) GTT (n = 10). (**E**) AUC for the GTT. (**F**) ITT (n = 10). (**G**) AUC for the ITT. (**H**) Triglycerides, (**I**) cholesterol, (**J**) HDL, and (**K**) LDL serum levels, (**H**–**K**): HFD n = 4, HFD NE n = 6, and HFD EE n = 10). (**L**) Percentage of infiltrating cell count and (**M**) infiltrated area in epididymal white adipose tissue determined by hematoxylin-eosin staining (n = 3). Bars represent the mean ± SEM. * *p* < 0.05, ** *p* < 0.01, and *** *p* < 0.001 vs. ND mice; ^#^ *p* < 0.05 HFD vs. HFD EE; ^&^ *p* < 0.05 HFD NE vs. HFD EE; ^###^ *p* < 0.001 HFD vs. HFD EE; ^&&&^ *p* < 0.001 HFD NE vs. HFD EE (two-way ANOVA followed by a Bonferroni post-hoc test (A–M)).

**Figure 5 ijms-25-06904-f005:**
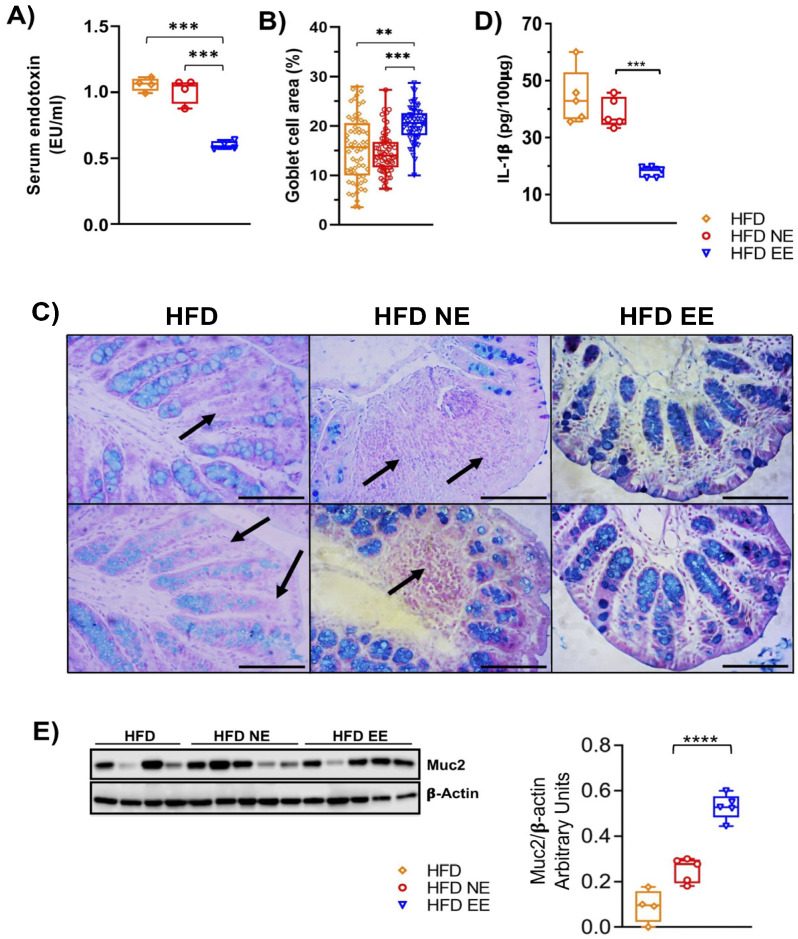
**Environmental enrichment increases the levels of Muc2 in the proximal colon of mice fed with high-fat diet.** Mice fed a HFD for 12 weeks in standard housing conditions (HFD) or maintained for an additional 12 weeks in standard housing (HFD NE) or EE (HFD EE) conditions were euthanized, and the proximal colon and serum from the experimental groups were collected. (**A**) Circulating lipopolysaccharide levels (n = 4). The proximal colon of three mice from the various conditions were fixed, embedded in paraffin, sliced, and stained with Alcian blue. (**B**) Percentage of the area occupied by goblet cells. (**C**) Representative photos of the proximal colon stained with Alcian blue for acidic mucin in the goblet cells. Arrows indicate the absence or decrease in goblet cells. Scale bar: 100 μm. (**D**) IL-1β levels in the proximal colon. (**E**) Muc2 protein levels in the proximal colon. Left panel: representative western blot of Muc2 in the experimental groups. Right panel: densitometric analysis of Muc2 protein levels normalized to β-actin values (HFD n = 4 and HFD NE and HFD EE n = 5). Data are mean ± S.E.M. ** *p* < 0.01, *** *p* < 0.001, and **** *p* < 0.0001 vs. HFD or HFD NE (unpaired two-tailed *t*-test).

**Figure 6 ijms-25-06904-f006:**
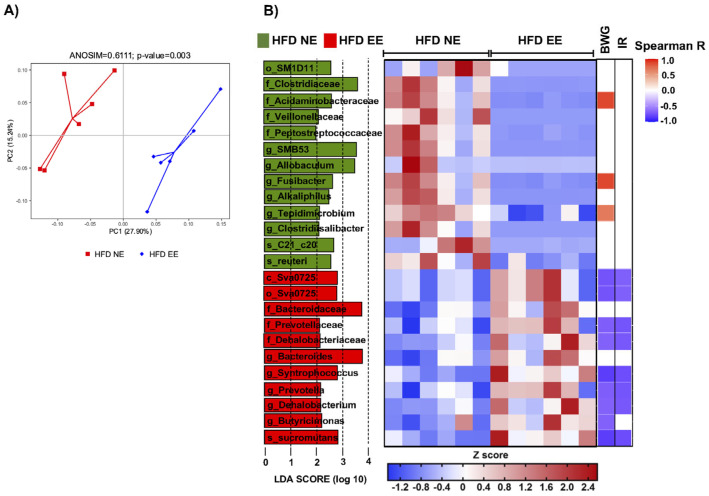
**Environmental enrichment regulates gut microbiota dysbiosis and increases specific intestinal taxa**. (**A**) Principal component analysis showing the beta diversity clustering of the gut microbiota from mice fed a HFD for 24 weeks and maintained in standard housing conditions (HFD NE) and from mice maintained in environmental enrichment (HFD EE). (**B**) Left: LEfSe comparing bacterial class, order, family, genus, and species between the HFD NE and HFD EE groups. Middle: heatmap showing the relative abundance of the taxa enriched in the HFD NE and HFD EE groups. Right: heatmap showing the Spearman correlation coefficient between body weight gain (BWG) and insulin resistance (IR) with several of the taxa enriched in the HFD NE and HFD EE groups. c_: class; o_: order; f_: family; g_: genus; s_: species. n = 6 from each group. See Appendix A for the taxonomic ranks of the bacteria.

## Data Availability

The sequencing data utilized in this study have been deposited under BioProject PRJNA1069107 into the NCBI database. Data are available only upon request to the corresponding author. Full availability only after article acceptance.

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
