# Peer review of "Environmental Enrichment Prevents Gut Dysbiosis Progression and Enhances Glucose Metabolism in High-Fat Diet-Induced Obese Mice"

_ijms, 2024, doi:10.3390/ijms25136904_

Round 1
Reviewer 1 Report
Comments and Suggestions for Authors
This study explores how the environment enrichment affects the microbiota and various aspects of metabolism in mice subjected to a high-fat diet. The results are interesting and make us think about the importance of physical activity and external stimuli to maintain a healthy state and open the door to addressing certain metabolic diseases with therapeutic tools based on probiotics or postbiotics that are modified in an enriched environment.
There are some points that need to be further developed before the paper can be published:
General comments
1. The results and discussion sections are too long. This animal model (HFD diet obese animals) is very well established and many of the results presented confirm those described previously. These sections should be shortened.
2. A deeper discussion is lacking about the cause of the metabolic changes observed in animals subjected to a HFD and in an enriched environment. The authors describe the situation, but do not discuss it in sufficient depth. For example, does the environmental enrichment influence the animal's physical activity? Can it be assumed that in an enriched environment where animals are subjected to more stimuli, they will move more? Could this be the cause of some of the observed results? These elements could be discussed in relation to published work on the relationship between physical activity and the microbiota, and in the case of the cognitive stimuli to which the mice are subjected.
Methods
3. The composition of the diets must be included.
4. Line 622. The authors indicate that "fecal samples are combined to homogenize the intestinal microbiota". We can understand that it involves collecting the samples from the 15 animals in a cage and mixing them. But it cannot be said that N=15, since only one sample is analyzed. This point and the N of all the figures in the manuscript must be reviewed.
5. Throughout the study, the statistical analysis should be reviewed. In the case of microbiota, the number of samples analyzed must be taken into account; and the number of animals in the case of the results in the histology section.
Results
6. Please check the Figure S1D legend. It is not understood because the authors refer to the two diets, ND and HFD. Also, the N is not clear. In the figure, what do the notations on the abscissa axis correspond to?
7. Figure 1. Please, check and indicate the N value in each case.
8. Line 185. Figure 2. Please check the legend. What samples were collected? Gut microbiota or gut content?
9. Line 133. It is suggested to begin a new paragraph in the sentence “Once our diet-induced ...”
10. Figure 3. Some of the subfigures repeat results from others. This is the case, for example, of figures 3.F and 3.G with respect to 3.A and 3.B.
11. Figure 3. It is recommended to use the same color code in the case of 3.D and 3.H
12. In section 2.2 there is repeated information. For example, lines 240-242 and 248-251. Please summarize the text.
13. Figure 4. The foot of the figure is excessively long. For example, there is no need to repeat the experimental design in detail.
14. Figure 5.B. It is indicated that this figure corresponds to “percentage of the area occupied by goblet cells”. Could you explain how it was measured? And also, what N was considered for the statistical analysis?
15. Figure 5.C. Representative photos of Figure 5.C-HFD-NE do not show a villous area. Could you show other samples where the changes in goblet cells are more visible?
16. The last paragraph of the results sections does not correspond to 2.3. section (... function of the intestinal epithelial barrier ...). These results should be separated in another section.
Discussion
17. Line 401. The number of significant figures must be consistent with the sensitivity of the analytical method used (the sensitivity of the balance, in this case).
18. The explanation they give by the authors about the differences with the results of previous works is not entirely consistent. For example, the authors find no differences between ND and HFD animals in some taxa for which there is a large literature, and they attribute this to the fact that the type of sequencing applied is different. The argument enters the realm of speculation.
19. Lines 487-489. The authors report that they see no differences in the tight junction protein levels between HFD-NE and HFD-EE mice. However, they suggest that EE improves intestinal barrier protection. Could you provide some more documented arguments to support this claim?
20. Lines 502-505. The authors make the following statement in the discussion “There is a necessity of housing model organisms, including rats and mice, in environments enriched to preserve physiological health and ensure the manifestation of innate behaviors”. What are the consequences of this statement in the studies that have been done on obesity in animal models?
Thank you
Reviewer 2 Report
Comments and Suggestions for Authors
Summary: The study delves into the therapeutic potential of Environmental Enrichment (EE) in addressing high-fat diet-induced metabolic syndrome. It reveals that EE intervention effectively halts dysbiosis progression, enhances intestinal barrier integrity, and restores glucose metabolism to normal levels. These findings underscore EE's promise as an anti-inflammatory strategy for managing obesity-related metabolic dysregulation.
Some comments for improving the manuscript:
1. In the abstract you write: Here, we investigated the therapeutic potential of Environmental Enrichment (EE) in mitigating high fat diet (HFD)-induced metabolic syndrome through modulation of gut microbiota composition.
It is not possible to state that EE improves HFD-induced metabolic syndrome through modulation of the intestinal microbiota, because you have not done an experiment using germ-free mice. The mechanisms of action responsible for the improvements observed could be other than the modulation of dysbiosis.
2. In the abstract: mention of the methods is completely missing.
3. In the Introduction: insert more concepts related to obesity, inflammation and the changes of the intestinal microbiota following a high-fat diet.
Explain the relationship between the alteration of the intestinal microbiota and the development of metabolic diseases.
Clarify how environmental enrichment can influence the intestinal microbiota and how this modulation can contribute to the management of obesity and associated dysmetabolic conditions.
Shorten it by eliminating concepts that digress from the objective of the study (for example from line 54 to 67) by focusing on what this reviewer has indicated above.
In the introduction of a scientific article it is preferable to focus on the presentation of the context, the importance of the problem and the objectives of the study rather than on the results obtained and future prospects. Results obtained and future prospects are best reserved for the conclusion section.
Usually at the end of the introduction the purpose of the study is indicated, which is missing here.
4. Results: this reviewer suggests changing the layout of the paragraphs to make the understanding of the results clearer and avoid repetitions, dividing them into:
- metabolic analysis on mice before changing the environmental conditions,
- metabolic analysis after inserting EE at the end of 24 weeks among the 4 experimental groups,
- analysis of the microbiota after EE at the end of 24 weeks among the 4 experimental groups,
- analysis among the 12 weeks HFD and 24 weeks HFD with and without EE,
- analysis of the intestinal barrier.
5. Line 133-138: should be moved to materials and methods.
6. Line 147-149: Why HFD NE mice have higher levels of HDL than ND mice? it should be the opposite.
7. Line 153-155: to be able to state that the adipose tissue had fewer infiltrating cells, immunohistochemistry should be performed, highlighting the macrophages and crown like structures. Why wasn't this done? this reviewer suggests doing so.
Furthermore, to be able to state that EE mitigates the damage caused by HFD on adipose tissue, the size of the adipocytes should be measured by evaluating their area or diameter. This reviewer suggests doing so.
8. Line 158-160: if out of three inflammatory markers only 1 changes significantly compared to the control, it cannot be said that EE has the ability to reduce obesity-induced inflammation in adipose tissue. But there is only one tendency to reduction.
9. Figure 3: “Environmental enrichment prevents the progression of gut microbiota dysbiosis induced by long term high fat diet consumption” it should be bold. Letters indicating graphs should also be bold.
10. Line 252: specify the acronym LDA.
11. Line 300: specify where the LPS is evaluated.
12. Figure 5: A: add "serum endotoxin EU/ml" to the y-axis.
13. Line 319-321: in the materials and methods it is indicated that levels of IL6, IL1beta, IL10 and TNF alpha are measured. Why is only IL1beta shown?
14. Line 330-354: it is a repetition of the analysis done and indicated by figure 3D, 3H. These results can be grouped together by specifying which group is being analysed.
This reviewer suggests combining all this analysis of the microbiota with the one done previously also to have conceptual continuity. Inserted in the paragraph “Environmental enrichment restores the function of the intestinal epithelial barrier in mice fed 292
with a high-fat diet” turns out to be out of context.
15. Discussion: too long and sometimes redundant with concepts repeated several times. This reviewer suggests shortening.
16. Line 633-634: specification of the time the mice were on EE and the different diets is missing.
17. Material and methods:
- Animals: How were the mice sacrificed? how many hours of fasting did they do?
- Basal glucose levels, glucose tolerance, and insulin resistance test: subcutaneous glucose injection? intraperitoneal? Were the mice fasting? if so how many hours?
- Lipids determination: what sample was used? serum? how many microliters?
You write: “Total cholesterol and triglycerides (TG) were quantified using an enzymatic colorimetric assay with the glucose oxidase test”; how can you measure triglyceride levels using a glucose oxidase test?
“4-aminophenazone” is it a typo? Clarify.
“Roche-Cobas C111 kit, Roche Diagnostic USA” there are many types of tests that can be done, specify the one used to evaluate cholesterol levels and the one used to evaluate triglyceride levels.
- Mice perfusion: how come the guys were anesthetized with ketamine and xylazine? these anesthetics could influence aspects of metabolism, the endocrine system and also have effects on adipose tissue...
why wasn't perfection achieved immediately after euthanasia?
What is the perfusion access route? ascending aorta? portal vein? Specify
- Tissue collection and analysis: why was CO2 inhalation or perfusion done if the mice had already been anesthetized and perfused?
- Total protein extraction and western blot assay: “Total protein was extracted from the proximal colon” and those of adipose tissue?
Specifications of secondary antibodies are missing.
- ELISA: interleukins have also been measured in adipose tissue. Enter it.
In the section on protein extraction only extraction from the colon was mentioned, omitting that it was also done in adipose tissue. Enter it.
- LPS quantification: define the type of sample used.
Comments on the Quality of English Languagethe English used seems adequate to these reviewer. Only a minor revision is required.
Round 2
Reviewer 2 Report
Comments and Suggestions for Authors
Dear authors, I have no further comments to make.